# Experimental Study: The Development of a Novel Treatment for Chemotherapy-Resistant Tongue Cancer with the Inhibition of the Pathological Periostin Splicing Variant 1-2 with Exon 21

**DOI:** 10.3390/cells13161341

**Published:** 2024-08-13

**Authors:** Shoji Ikebe, Nobutaka Koibuchi, Kana Shibata, Fumihiro Sanada, Hideo Shimizu, Toshihiko Takenobu, Yoshiaki Taniyama

**Affiliations:** 1Graduate School of Dentistry (Second Department of Oral and Maxillofacial Surgery), Osaka Dental University, Hirakata 573-1121, Japan; ikebe-s@cc.osaka-dent.ac.jp; 2Department of Advanced Molecular Therapy, Graduate School of Medicine, Faculty of Medicine, Osaka University, Suita 565-0871, Japan; koibuchi@cgt.med.osaka-u.ac.jp (N.K.); shibata@periotherapia.co.jp (K.S.); 3Department of Clinical Gene Therapy, Graduate School of Medicine, Faculty of Medicine, Osaka University, Suita 565-0871, Japan; sanada@cgt.med.osaka-u.ac.jp; 4Department of Internal Medicine, Osaka Dental University, Hirakata 573-1121, Japan; taropapa@mx6.nisiq.net; 5Second Department of Oral and Maxillofacial Surgery, Osaka Dental University, Hirakata 573-1121, Japan; takenobu-t@cc.osaka-dent.ac.jp

**Keywords:** periostin, POSTN, POSTN exon 21, splicing variant, tongue cancer, tongue squamous cell carcinoma, chemoresistance

## Abstract

Tongue squamous cell carcinoma (TSCC) occurs frequently in the oral cavity, and because of its high proliferative and metastatic potential, it is necessary to develop a novel treatment for it. We have reported the importance of the inhibition of the periostin (POSTN) pathological splicing variant, including exon 21 (PN1-2), in various malignancies, but its influence is unclear in tongue cancer. In this study, we investigated the potential of POSTN exon 21-specific neutralizing antibody (PN21-Ab) as a novel treatment for TSCC. Human *PN2* was transfected into the human TSCC (HSC-3) and cultured under stress, and PN2 was found to increase cell viability. PN2 induced chemotherapy resistance in HSC-3 via the phosphorylation of the cell survival signal Akt. In tissues from human TSCC and primary tumors of an HSC-3 xenograft model, PN1-2 was expressed in the tumor stroma, mainly from fibroblasts. The intensity of *PN1-2* mRNA expression was positively correlated with malignancy. In the HSC-3 xenograft model, CDDP and PN21-Ab promoted CDPP’s inhibition of tumor growth. These results suggest that POSTN exon 21 may be a biomarker for tongue cancer and that PN21-Ab may be a novel treatment for chemotherapy-resistant tongue cancer. The treatment points towards important innovations for TSCC, but many more studies are needed to extrapolate the results.

## 1. Introduction

Head and neck squamous cell carcinoma (HNSCC) is one of the most prevalent cancers; it has an annual morbidity of more than 500,000 cases worldwide [1]. Tongue cancer (TSCC, or tongue squamous cell carcinoma) is the most common HNSCC and is characterized by high proliferative and infiltrative ability, a high lymph node metastasis rate, and increasing incidence year by year [2,3,4]. Generally, the primary treatment of TSCC is selected based on tumor stage, with glossectomy remaining the best option. For cases in which surgical resection is difficult due to the tumor’s stage of progression or as a form of postoperative therapy, radiotherapy or chemotherapy, or their combination, is recommended [5,6,7]. Chemotherapy resistance is a critical issue encountered in the treatment of cancers. Because cancer cells quickly develop resistance to forms of first-line chemotherapy, such as 5-fluorouracil (5-FU) and cisplatin (and target drugs), clinical outcomes are unsatisfactory in the majority of patients; however, the reasons for this remain unclear. Researchers have continued to seek alternative treatments, but they have not been sufficient to ameliorate the current status of TSCC. Therefore, it is important to identify the mechanism behind this drug resistance, as overcoming this obstacle could lead to the development of effective treatments [8].

On the other hand, periostin (POSTN), also termed osteoblast-specific factor 2, is a matricellular protein known to function in osteology, tissue repair, oncology, cardiovascular and respiratory systems, and various inflammatory settings. However, POSTN performs physiological functions and has been reported to be involved in various intractable diseases [9]. There are also reports that POSTN promotes chemotherapy resistance in various malignancies such as breast, lung, and pancreatic cancer [10,11,12]. POSTN has four main splicing variants: PN1 (full-length POSTN), PN2 (POSTN lacking exon 17), PN3 (POSTN lacking exon 21), and PN4 (POSTN lacking exons 17 and 21) (Figure 1) [13]. We call PN4 the physiological POSTN and PN1-3 the pathological POSTN. In particular, we have previously reported the importance of inhibiting the pathological POSTN splicing variant with exon 21 (PN1-2) in chemotherapy-resistant breast cancer, malignant melanoma, diabetic retinopathy, inflammatory bowel disease, atherosclerosis, aortic aneurysm, etc. [14,15,16].

Kudo et al. reported that POSTN plays an important role in the proliferation and invasion of HNSCC, and Shao et al. reported that POSTN isoform-lacking exon 17 is a factor that promotes tumor progression in HNSCC [17,18]. However, there are no reports, to the best of our knowledge, of the tissue expression of *PN1-2* mRNA in TSCC and the inhibitory effect of POSTN exon 21-specific neutralizing antibody (PN21-Ab) on the progression of cancer. 

In this study, we investigated the effect of PN21-Ab with or without conventional chemotherapy on TSCC cells in vivo, and we also investigated its mechanism in vitro.

## 2. Materials and Methods

### 2.1. Ethical Statement

All the experimental methods described herein were conducted with the approval of the Medical Ethics Committee of Osaka Dental University (Approval No. 111328).

All animal experiments were performed in accordance with the protocols approved by the Animal Ethics Committee of Osaka University (Approval No. 04-101-007). 

### 2.2. Cell Cultures

In this study, we performed in vitro and in vivo studies using *human* tongue squamous cell carcinoma (TSCC) cell line HSC-3. HSC-3 cells were obtained from the Cell Resource Center for Biomedical Research, Institute of Development, Aging and Cancer, Tohoku University (Miyagi, Japan). The HSC-3 cells were maintained in RPMI-1640 with L-Glutamine and Phenol Red (FUJIFILM Wako Pure Chemical Corporation, Osaka, Japan), supplemented with 10% (*v*/*v*) fetal bovine serum (FBS) (Biowest, Nuaille, France) and 1% penicillin/streptomycin (Nacalai Tesque, Kyoto, Japan) under 5% CO_2_ at 37 °C.

### 2.3. Anti-Human POSTN Antibody

In order to raise the mouse monoclonal antibody against exon 21 of human POSTN, the exon 21 peptide was synthesized at Oriental Yeast Co., Ltd (Tokyo, Japan). The antibody was generated in immunized mice, as described previously [15].

### 2.4. In Situ Hybridization

We evaluated the expression patterns of all *POSTN* splicing variants (*PN1-4*) and only *PN1-2* mRNA on tissues using human tongue cancer clinical samples. We selected TSCC as the sample and obtained tissue arrays of TSCC from TissueArray.Com (Derwood, MD, USA). Specifically, we stained *PN1-4* mRNA via in situ hybridization (ISH), RNAscope^TM^ (Advanced Cell Diagnostics, Newark, CA, USA), and *PN1-2* mRNA using ISH, BaseScope^TM^ (Advanced Cell Diagnostics, Newark, CA, USA) on normal human tongue and TSCC tissues. These procedures were performed according to a manual from Advantech Co., Ltd. (Tokyo, Japan).

### 2.5. MTS Cell Proliferation Assay

We plated HSC-3 cells in 96-well plates at a density of 2 × 10^4^ cells/well and maintained them for 16 h at 37 °C. Next, because the *human* PN2 protein is not commercially available, we transfected the *PN2* gene or Venus into the HSC-3 cells using Lipofectamine^®^3000 Reagent (Thermo Fisher Scientific Inc., Tokyo, Japan) and maintained them for 6 h at 37 °C. After the medium was changed to be serum-free and was maintained for 5 days at 37 °C, we performed a proliferation assay using a CellTiter96^®^AQueous One Solution Cell Proliferation Assay (Promega, Madison, WI, USA). Additionally, in another medium, we added the anti-cancer agent Cisplatin (CDDP) in different proportions (1, 2, and 5 µM) when we changed the medium to being serum-free. After maintaining that medium for 2 days at 37 °C, we performed an MTS assay in the same way. The CDDP was obtained from AdipoGen Life Sciences (San Diego, CA, USA).

### 2.6. Western Blot Analysis

We plated HSC-3 cells in 96-well plates at a density of 2 × 10^4^ cells/well and maintained them for 24 h at 37 °C. After we transfected the *PN2* gene or Venus into HSC-3 cells and maintained them for 6 h at 37 °C, the medium was changed to be serum-free. In addition, CDDP (5 µM) was added in some cases. The total proteins were extracted using an EzRIPA Lysis kit (ATTO, Tokyo, Japan), and their concentrations were determined via a DC protein assay (BIO-RAD, Tokyo, Japan). We adjusted the protein concentration, performed electrophoresis, and blotted onto PVDF membranes. After blocking with Ez Block Chemi (ATTO, Tokyo, Japan) for 1 h at room temperature, the membranes were incubated with Phospho-Akt Antibody (#9271; Cell Signaling Technology, Inc: Danvers, MA, US) and Akt Antibody (#9272) at 4 °C overnight. Following washing with TBST three times at room temperature, each time for 10 min, they were incubated with anti-rabbit IgG HRP-linked antibody (#7074; Cell Signaling Technology, Inc.) for 1 h and then washed again. Protein bands were visualized with an enhanced Amersham™ ECL™ Prime (#RPN2232; Cytiva, Tokyo, Japan). GAPDH (#97116; Cell Signaling Technology, Inc.) was used as the internal control. ImageQuant™ 4000 mini (Cytiva, Tokyo, Japan) was used for the analysis of total-Akt and phosphor-Akt expression. The phosphorylation of Akt in PN2-transfected HSC-3 was compared and examined between CDDP treatment and non-treatment.

### 2.7. Xenograft Assay in Nude Mice

We used female BALB/c nude mice obtained from the Jaxon Laboratory (ME, USA) in this experiment. We suspended 5 × 10^6^ cells from HSC-3 in 50% matrigel and transplanted them subcutaneously into the back of 7-week-old female BALB/c nude mice (each animal number = 4–5). When the tumor volume reached 30–40 mm^3^, CDDP (4 mg/kg) was injected intraperitoneally for 5 consecutive days. In some experiments, PN21-Ab (20 mg/kg) was injected intravenously twice weekly in combination with CDDP. Additionally, we removed primary tumors from the CDDP- and PN21-naïve HSC-3 xenograft model and fixed them with 4% paraformaldehyde. The specimens were embedded in paraffin, cut into 5 µm thick sections, and stained *PN1-4* mRNA using ISH, RNAscope^TM^ and *PN1-2* using ISH, BaseScope^TM^. Proves were used for hybridization: *POSTN* (#418581, 855401). We evaluated the expression patterns of *PN1-4* and *PN1-2* in primary tumor tissues of a xenograft model. One week after the primary tumor volume in the HSC-3 xenograft model reached 30–40 mm^3^, we measured the tumor volume for the group with no treatment, with CDDP alone, with PN21-Ab alone, and CDDP with PN21-Ab. Tumor volume (mm^3^) was calculated as 1/2 × width (mm) × length (mm), as described previously [19]. We evaluated the effects of CDDP with or without PN21-Ab in combination on the growth of HSC-3. 

### 2.8. Statistical Analysis

The results of our statistical analysis are presented as means ± SD. The Mann–Whitney test was performed to compare multiple treatment groups. For the statistical analysis of the change in expression in the two groups, the Wilcoxon signed-rank test was performed.

## 3. Results

### 3.1. Expression Pattern of PN1-4 and PN1-2 in Human TSCC

We performed ISH (RNAscope^TM^ and BaseScope^TM^) on a normal *human* tongue sample and two *human* TSCC samples (grade 1 and grade 2) and stained both *PN1-4* and *PN1-2* mRNA (Figure 2). To examine the histological expression of *PN1-4* and *PN1-2* mRNA, HE staining was performed on all tissues for comparison. No expression of *PN*1-4 or *PN*1-2 mRNA was observed in normal *human* tongue tissue. In *human* TSCC tissue, both samples showed expression of *PN1-4* and *PN1-2* mRNA in the tumor stroma. In addition, both *PN1-4* and *PN1-2* mRNA were more strongly expressed in the tumor stroma of human TSCC grade 2 compared to grade 1. The strong expression of *PN1-2* mRNA in histopathologically high-grade types suggests that POSTN exon 21 may be a marker that correlates with the grade of tongue cancer.

### 3.2. Effect of PN2 on Tongue Cancer Cell Viability

We cultured *PN2* transgenic HSC-3 under conditions of growth stress and evaluated their cell counts. When *PN2*-transfected HSC-3 was cultured under serum-free conditions, its cell counts were significantly higher than that of the control (Figure 3A). Therefore, we initially thought that PN2 inhibited the death of tongue cancer cells. In addition, when CDDP was administered to *PN2*-transfected HSC-3 in serum-free culture, there was no significant difference in cell counts between those without CDDP and those with CDDP (in proportions of 1 µM and 2 µM) compared to the control. However, the number of cells was significantly higher with the administration of CDDP 5 µM, and the number of surviving cells increased as the concentration of the anticancer drug increased (Figure 3B). This suggests that PN2 may confer chemotherapy resistance to HSC-3 alongside improving the survival of tongue cancer cells.

### 3.3. Activation of Akt by PN2 in Tongue Cancer Cells

We focused on the phosphorylation of Akt by PN2 to support the hypothesis that PN2 activates survival signals in tongue cancer cells. The phospho-Akt expression in *PN2* transgenic HSC-3 in serum-free culture was compared and validated with and without CDDP using Western blotting. Phospho-Akt was found to be significantly elevated in PN2-transfected HSC-3 in serum-free culture compared with a Venus vector (Figure 4). In Venus-transfected HSC-3, phospho-Akt was decreased in CDDP-treated HSC-3 compared to non-treated HSC-3. On the other hand, phospho-Akt was increased in CDDP-treated PN2-transfected HSC-3 compared with Venus vectors (Figure 5). 

### 3.4. Expression Pattern of PN1-4 and PN1-2 in an HSC-3 Xenograft Model

We have succeeded for the first time in specifically staining splicing variants of POSTN mRNA, especially pathological splicing variants including exon 21, on the pathology of the HSC-3 xenograft model primary tumor. *PN1-4* and *PN1-2* were found to be expressed primarily in the tumor stroma of the HSC-3 xenograft model. Among them, *PN1-2* and *PN1-4* were expressed mainly in fibroblasts in the stroma surrounding cancer spores (Figure 6).

### 3.5. Effect of PN21-Ab

The growth of primary tumors in the HSC3 xenograft model treated with PN21-Ab alone was inhibited compared to untreated tumors (Figure 7A). Additionally, the HSC3 xenograft model’s primary tumor volume was also compared between no treatment, CDDP treatment alone, and CDDP and PN21-Ab combined treatment. In individuals treated with CDDP alone, primary tumor growth was of course suppressed compared to non-treated individuals. However, the combination of CDDP and PN21-Ab markedly suppressed tumor growth compared to individuals treated with CDDP alone (Figure 7B). The results suggest that the inhibitory effect of CDDP on tumor growth was significantly enhanced by its combination with PN21-Ab.

## 4. Discussion

As previously stated, TSCC is one of the most frequently diagnosed malignant tumors in the oral cavity [20,21]. Because of its high metastatic and proliferative ability, TSCC represents a significant threat to *human* health worldwide [22,23]. Over the last few decades, progress has been made in the therapeutic management of TSCC in the form of surgery, chemotherapy, and radiotherapy; however, the five-year survival rate of TSCC patients is still less than 50% [22,24]. Therefore, there is an urgent need for researchers to comprehend the genetic and molecular mechanisms that underlie carcinogenesis and to develop clinically relevant biomarkers and novel treatments [25]. With this in mind, we focused on POSTN splicing variants, specifically PN1-2, in our pursuit of a novel treatment for tongue cancer. 

POSTN has been demonstrated to control various biological aspects of tumor cells, including proliferation, invasion, survival, angiogenesis, metastasis, and resistance to chemotherapy [26,27,28]. Moreover, POSTN plays a pivotal role in the remodeling of various tumor microenvironments, including the cancer stem cell niche, the perivascular niche, the premetastatic niche, and the immunosuppressive microenvironment [29]. Past reports have also suggested that POSTN with exon 21 has a clear role in supporting the development of cancer [14].

In this study, *PN2*-transfected HSC-3 in vitro exhibited a significant increase in cell survival in serum-free conditions and when CDDP was administered. This suggests that PN2 augmented the cellular survival of TSCC cells under the conditions of growth stress. In addition, phosphor-Akt was significantly increased in *PN2*-transfected HSC-3 treated with CDDP. Thus, PN2 was thought to activate survival signals in tongue cancer cells via Akt activation. According to previous reports, POSTN splicing variants are abundantly secreted and can bind to specific integrin receptors and activate the Akt/PKB pathway via the phosphorylation of the focal adhesion kinase (FAK) and phosphoinositide 3-kinase (PI3K) signaling pathways. This pathway is a downstream of integrin signaling and promotes cell migration and proliferation [30]. In colon cancer, POSTN has been shown to promote metastatic growth through augmenting cell survival via the Akt/PKB pathway [31]. In addition, Kudo et al. reported that POSTN-induced tumor lymphangiogenesis in HNSCC is mediated by Akt activity [17]. Other reports have suggested that AKT activation is a common molecular feature of *human* malignancies that occurs in a variety of cancers and is even associated with drug resistance in cancer cells [32]. Although not explored in detail in this study, as reinforced by previous findings, PN2 may confer chemotherapy resistance to tongue cancer cells via Akt activation as a cell survival signal, promoting the proliferation, invasion, and metastatic potential of TSCC.

Furthermore, for the first time, and using ISH, we succeeded in analyzing the patterns of *PN1-4* and *PN1-2*’s histological expression by staining *human* TSCC tissues and primary tumor tissues taken from a nude mouse HSC-3 xenograft model. No expression of either *PN1* or *PN2* was observed on normal *human* tongue tissue. In many cancers, POSTN is mostly found in the cancer stroma [33,34,35]. Several studies have found that the overexpression of POSTN in tumor stroma is associated with more aggressive tumors, advanced stage or poor prognosis, and shorter overall survival in various cancers [29]. In invasive ductal breast carcinoma, higher POSTN expression levels in cancer-associated fibroblasts (CAFs) have been associated with higher tumor cell grade and shorter overall survival, suggesting that POSTN secreted by fibroblasts may be a marker of breast cancer progression [36]. POSTN has also been reported to be expressed on CAFs in colorectal and gastric cancer [37,38]. In this experiment, we observed PN1-4 and PN1-2 expression in the tumor stroma on both *human* TSCC and a primary tumor from a nude mouse HSC-3 xenograft model. In addition, PN1-2 was expressed in the primary tissues of the transplantation model, even though PN2 was almost not expressed in HSC-3. This suggests that PN1-2 is induced in the host (*mice*) through some mechanism. In both tissues, *PN1-2* mRNA was expressed primarily from stromal fibroblasts. It is possible that PN1-2 is expressed from CAFs. This is consistent with a report that POSTN is mainly observed in CAFs according to RNA-seq data obtained from the GEO database concerning single-cell transcriptomes from HNSCC patients [18]. These results suggest that the primary cells that produce and secrete POSTN are fibroblasts in the cancer stroma, i.e., CAFs, which have the characteristics of myofibroblasts. Additionally, these findings suggest that POSTN produced by CAFs may constitute a growth-supportive microenvironment for TSCC. In *human* TSCC tissues, *PN1-2* mRNA was strongly expressed in less differentiated tissues, suggesting that POSTN exon 21 may be a marker of malignant potential.

In terms of economic considerations, the treatment of pathological periostin antibodies once every two weeks during anticancer drug treatment, for example, 12 times in 6 months, will increase the burden. Neutralizing antibodies that have been approved in combination with anticancer drugs in various cancers, such as immune checkpoint inhibitors in breast cancer, exist and are considered acceptable.

In summary, we hypothesized that pathological POSTN with exon 21 expressed in stromal fibroblasts may confer chemotherapy resistance to tongue cancer cells via Akt activation and activate cell survival signals (Figure 8). In light of the above, we evaluated the effects of the intravenous administration of PN21-Ab in a CDDP-treated mouse model of tongue cancer. Tumor growth in the primary tumor was markedly suppressed compared with that without PN21-Ab. The results suggest the potential of a novel form of therapy for targeting POSTN exon 21 in tongue cancer that has acquired resistance to chemotherapy with CDDP. Fujikawa et al. reported that in triple-negative breast cancer, PN21-Ab suppressed tumor cell growth, accompanied by decreased M2 tumor-associated macrophage polarization and a decrease in the number of tumor vessels [15]. The mechanism causing POSTN to splice with exon 21 in tongue cancer is unknown; that said, the targeting of pathological POSTN splicing variants with PN21-Ab might offer a safe and effective strategy for the treatment of tongue cancer. 

As mentioned above, the literature reports the involvement of POSTN in head and neck cancer metastases. Our future work will be to evaluate the association of POSTN exon 21 with the invasion and metastatic potential of tongue cancer and to test the efficacy of a POSTN exon 21-specific neutralizing antibody as an inhibitor of lymph node metastasis.

## 5. Conclusions

POSTN exon 21 is a potential a biomarker of tongue cancer, and PN21-Ab could be the basis of a novel therapy for chemotherapy-resistant tongue cancer. The treatment points towards important innovation for tongue cancer, but many more studies are needed to extrapolate the results.

## 6. Patents

The patent of Ex21 antibody belongs to Osaka University, Periotherapia Co., which has the priority negotiation right.

## Figures and Tables

**Figure 1 cells-13-01341-f001:**
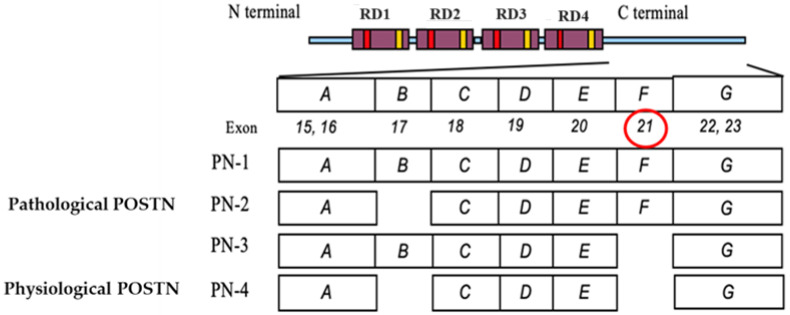
The N-terminus of POSTN has four repeat domains (FAS1). The C-terminal region (exons 15–23) centered on PN1-4 undergoes alternative splicing. Pathological POSTN splicing variants include exons 17 and 21 (PN1-3).

**Figure 2 cells-13-01341-f002:**
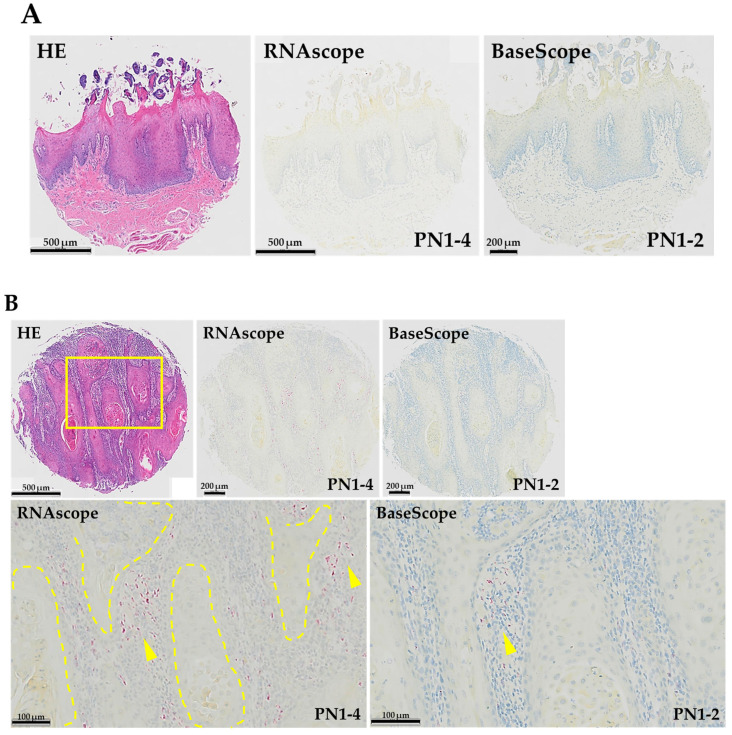
Hematoxylin–eosin (HE)-stained images and expression patterns of *PN1-4* and *PN1-2* mRNA using ISH (RNAscope^TM^, BaseScope^TM^) in each tissue. (**A**) Normal *human* tongue tissue. (**B**) *Human* TSCC (grade-1) tissue. (**C**) *Human* TSCC (grade-2) tissue. In (**B**,**C**), the lower images (RNAscope^TM^, BaseScope^TM^) are magnified views of the yellow box in the upper image (HE). The tumor parenchyma is surrounded by yellow dashed lines. *PN*1-4 and *PN*1-2 mRNA are stained red (as indicated by the yellow arrows).

**Figure 3 cells-13-01341-f003:**
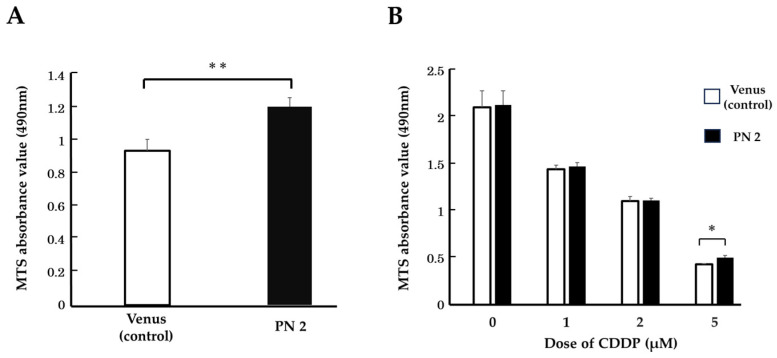
Impact of PN2 on cell death in HSC-3. (**A**) Number of HSC-3 cells transfected with *PN2* in serum-free culture (*n* = 8, **; *p* < 0.01 vs. control). (**B**) Number of *PN2*-transfected HSC-3 cells in serum-free culture with cisplatin treatment (*n* = 8, *; *p* < 0.05 vs. control).

**Figure 4 cells-13-01341-f004:**
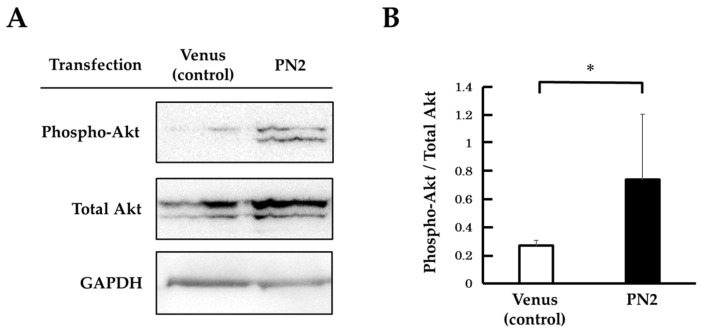
Protein expression of phospho-Akt in *PN2* transgenic HSC-3 (serum-free culture). (**A**) Expression levels of phospho-Akt and total Akt, as measured by Western blotting. (**B**) Histograms indicating the relative expression levels of phospho-Akt/total Akt (*; *p* < 0.05 vs. Venus).

**Figure 5 cells-13-01341-f005:**
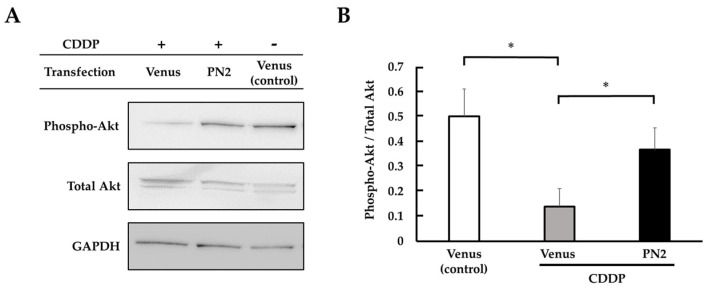
Protein expression of phospho-Akt in *PN2*-transfected HSC-3 (with CDDP administration). (**A**) Expression level of phospho-Akt and total Akt, as detected by Western blotting. (**B**) Histograms indicating the relative expression levels of phospho-Akt/total Akt. CDDP alone significantly decreased phospho-Akt/total Akt compared to Venus (*; *p* < 0.05 vs. control), and CDDP with *PN2* transfection significantly increased phospho-Akt/total Akt compared to CDDP alone (*; *p* < 0.05 vs. Venus).

**Figure 6 cells-13-01341-f006:**
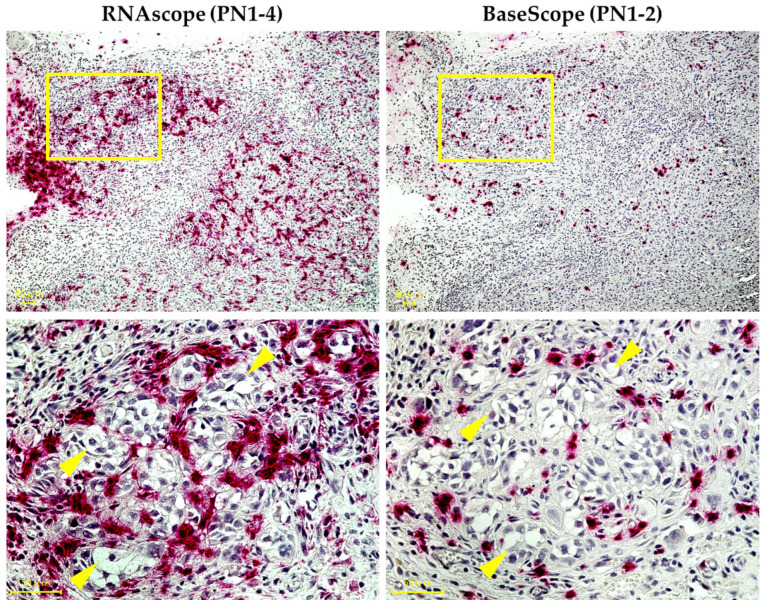
Expression patterns of *PN1-4* and *PN1-2* mRNA using ISH (RNAscope^TM^, BaseScope^TM^) in tissues of HSC-3 xenograft model primary tumors. The lower images are magnified views of the yellow box in the upper images. *PN1-4* and *PN1-2* are stained red. Cancer cells (cancer foci) are indicated by yellow arrows. The yellow scale bar indicates 50 µm.

**Figure 7 cells-13-01341-f007:**
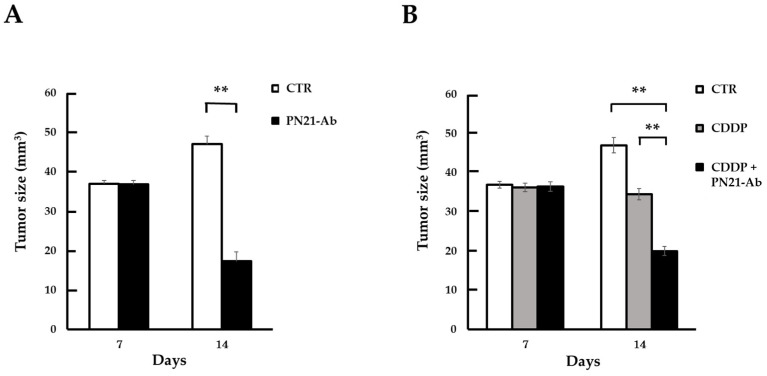
(**A**) Tumor size of primary tumors in an HSC-3 xenograft model treated with PN21-Ab; PN21-Ab significantly decreased tumor size at 14 days (*n* = 5, **; *p* < 0.01 vs. control). (**B**) Tumor size of primary tumor in HSC-3 xenograft model treated with CDDP and PN21-Ab; CDDP significantly decreased tumor size compared to the control, and CDDP with PN21-Ab significantly decreased it compared to CDDP alone (*n* = 4, **; *p* < 0.01 vs. control or CDDP with PN21-Ab). Tumor size (mm^3^) was calculated as 1/2 × width (mm) × length (mm).

**Figure 8 cells-13-01341-f008:**
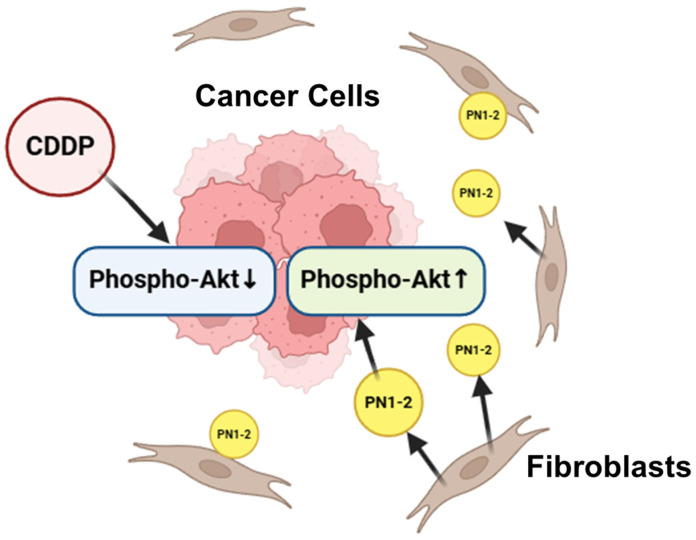
Possible mechanism of PN1-2 expression. PN1-2 is expressed primarily in stromal fibroblasts. Of course, the anticancer effect of CDDP decreases phospho-Akt in tongue cancer cells. However, it is thought that PN1-2 expressed from host stromal fibroblasts actually increases phospho-Akt.

## Data Availability

The data that support the findings of this study are available from the corresponding author, Y. Taniyama, upon reasonable request.

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
