# Peer review of "Experimental Study: The Development of a Novel Treatment for Chemotherapy-Resistant Tongue Cancer with the Inhibition of the Pathological Periostin Splicing Variant 1-2 with Exon 21"

_cells, 2024, doi:10.3390/cells13161341_

Round 1

Reviewer 1 Report

Comments and Suggestions for Authors

The manuscript is of excellent quality and addresses a topic of great relevance to public health and dentistry. I only suggest a few additions/changes to the text to make it even more complete:

- In the title, I consider it relevant to include that this is an experimental study;

- necessary both in the summary and in the conclusions to relativize the potential for extrapolation of the study to in vivo models and with human beings. "The treatment points towards important innovation for tongue CA, but many more studies are needed to extrapolate the results"

- It is necessary to discuss the cost of the method - is it clinically viable? What would be your limits?

I congratulate the authors on the excellent product presented!

Author Response

We appreciate your comments and criticisms of our manuscript. We are very happy to hear that “The manuscript is of excellent quality and addresses a topic of great relevance to public health and dentistry. I only suggest a few additions/changes to the text to make it even more complete”. We have improved our manuscript in compliance with your suggestions.

Comment1: In the title, I consider it relevant to include that this is an experimental study;

Response1: We changed following sentence in title according to the comment.

This is an Experimental Study; Development of a Novel Treatment for Chemotherapy-Resistant Tongue Cancer with the Inhibition of the Pathological Periostin Splicing Variant with Exon 21

Commnet2: - necessary both in the summary and in the conclusions to relativize the potential for extrapolation of the study to in vivo models and with human beings. "The treatment points towards important innovation for tongue CA, but many more studies are needed to extrapolate the results"

Response2: We added “The treatment points towards important innovation for TSCC, but many more studies are needed to extrapolate the results.” in both in the summary and in the conclusions. Because the letter number increased, we changed some sentence as follows.

Abstract: Tongue cancer Tongue squamous cell carcinoma (TSCC) occurs frequently in the oral cavity, and because of its high proliferative and metastatic potential, it is necessary to analyze its genetic and molecular mechanisms and develop a novel form of treatment. We have reported the importance of inhibition of periostin (POSTN) pathological splicing variant including exon 21 (PN1-2) in various malignancies, but its influence is unclear in tongue cancer. In this study, we investigated the potential of POSTN exon 21-specific neutralizing antibody (PN21-Ab) as a novel treatment for TSCC tongue cancer. Human PN2 was transfected into the human oral squamous cell carcinoma cell TSCC (HSC-3) and cultured under stress, and PN2 was found to increase cell viability. PN2 induced chemotherapy resistance in HSC-3 via phosphorylation of the cell survival signal Akt. In tissues from human TSCC and primary tumors of an HSC-3 xenograft model, PN1-2 was expressed in the tumor stroma, mainly from fibroblasts. The intensity of PN1-2 expression was positively correlated with malignancy. In the HSC-3 xenograft model, CDDP and PN21-Ab promoted CDPP’s inhibition of tumor growth. These results suggest that POSTN exon 21 may be a biomarker for tongue cancer and PN21-Ab may be a novel treatment for chemotherapy-resistant tongue cancer. The treatment points towards important innovation for TSCC, but many more studies are needed to extrapolate the results.

Commnet3: It is necessary to discuss the cost of the method - is it clinically viable? What would be your limits?

Response2: We added following sentences in Discussion.

In terms of economic considerations, the treatment of pathological periostin antibodies once every two weeks during anticancer drug treatment, for example, 12 times in 6 months will increase the burden. Neutralizing antibodies that have been approved in combination with anticancer drugs in various cancers, such as immune checkpoint inhibitors in breast cancer, exist and are considered acceptable.

Reviewer 2 Report

Comments and Suggestions for Authors

In this study, we investigated the effect of PN21-Ab with or without conventional chemotherapy on TSCC cells in vivo, and we also investigated its mechanism in vitro

·       It has an appropriate structure, with a clear introduction, a well-developed body, and a definite conclusion The authors have correctly summarized the study The topic is relevant

·       The authors focus on the POSTN splicing variants, specifically PN1-2, in the search for a novel treatment for tongue cancer. The title does not reflect the type of study; it should be mentioned.

·       The major sections are connected, and the relationships between them are clearly expressed .All major sections (headings and subheadings) are related to the topic and contribute to answering the task or question.

·       the methodological description should be improved.

All animal experiments were performed in accordance with the protocols approved    Xenograft assay in nude mice. More data should be provided regarding the animal experimental part, such as the number of animals used….

·       The conclusions answer the research questions. POSTN exon 21 is a potential a biomarker of tongue cancer, and PN21-Ab could be the basis of a novel therapy for chemotherapy-resistant tongue cancer.

·     references are appropriate

Author Response

We appreciate your comments and criticisms of our manuscript. We are very happy to hear that “ It has an appropriate structure, with a clear introduction, a well-developed body, and a definite conclusion The authors have correctly summarized the study The topic is relevant”. We have improved our manuscript in compliance with your suggestions.

Comment1: The authors focus on the POSTN splicing variants, specifically PN1-2, in the search for a novel treatment for tongue cancer. The title does not reflect the type of study; it should be mentioned.

Response1: We added 1-2 in the title according to the comment. We added This is an Experimental Study; ” according to Reviewer 1 comment.

This is an Experimental Study; Development of a Novel Treatment for Chemotherapy-Resistant Tongue Cancer with the Inhibition of the Pathological Periostin Splicing Variant 1-2 with Exon 21

Comment2:  The major sections are connected, and the relationships between them are clearly expressed .All major sections (headings and subheadings) are related to the topic and contribute to answering the task or question.

Response2: We appreciate the evaluation. 

Comment3:  the methodological description should be improved. All animal experiments were performed in accordance with the protocols approved    Xenograft assay in nude mice. More data should be provided regarding the animal experimental part, such as the number of animals used….

Response3: We added (each animal number=4-5) in 2.7. Xenograft assay in nude mice 

 Comment4:  The conclusions answer the research questions. POSTN exon 21 is a potential a biomarker of tongue cancer, and PN21-Ab could be the basis of a novel therapy for chemotherapy-resistant tongue cancer.

Response4:  We appreciate the evaluation. 

Comment5:  references are appropriate

Response5:  We appreciate the evaluation.